# Non-Deep Physiological Dormancy in Seed and Germination Requirements of *Lysimachia coreana* Nakai

**Saeng Geul Baek** [1], **Jin Hyun Im** [2], **Myeong Ja Kwak** [1], **Cho Hee Park** [2], **Mi Hyun Lee** [2], **Chae Sun Na** [2,*] and **Su Young Woo** [1,*]

1   Department of Environmental Horticulture, University of Seoul, Seoul 02504, Korea; bsg1175@naver.com (S.G.B.); na8349@uos.ac.kr (M.J.K.)
2   Seed Conservation Research Division, National Arboretum Baekdudaegan, Gyeongsangbuk-do 36209, Korea; jhim04@koagi.or.kr (J.H.I.); epurerbeau@koagi.or.kr (C.H.P.); hyun3176@koagi.or.kr (M.H.L.)
*   Correspondence: chaesun.na@koagi.or.kr (C.S.N.); wsy@uos.ac.kr (S.Y.W.); Tel.: +82-54-679-2769 (C.S.N.); +82-10-3802-5242 (S.Y.W.)

**Abstract:** This study aimed to determine the type of seed dormancy and to identify a suitable method of dormancy-breaking for an efficient seed viability test of *Lysimachia coreana* Nakai. To confirm the effect of gibberellic acid ($GA_3$) on seed germination at different temperatures, germination tests were conducted at 5, 15, 20, 25, 20/10, and 25/15 °C (12/12 h, light/dark), using 1% agar with 100, 250, and 500 mg·$L^{-1}$ $GA_3$. Seeds were also stratified at 5 and 25/15 °C for 6 and 9 weeks, respectively, and then germinated at the same temperature. Seeds treated with $GA_3$ demonstrated an increased germination rate (GR) at all temperatures except 5 °C. The highest GR was 82.0% at 25/15 °C and 250 mg·$L^{-1}$ $GA_3$ (4.8 times higher than the control (14.0%)). Additionally, GR increased after cold stratification, whereas seeds did not germinate after warm stratification at all temperatures. After cold stratification, the highest GR was 56.0% at 25/15 °C, which was lower than the GR observed after $GA_3$ treatment. We hypothesized that *L. coreana* seeds have a non-deep physiological dormancy and concluded that 250 mg·$L^{-1}$ $GA_3$ treatment is more effective than cold stratification (9 weeks) for *L. coreana* seed-dormancy-breaking.

**Keywords:** cold stratification; gibberellic acid ($GA_3$); germination rate (GR); *Lysimachia*; seed dormancy; warm stratification

## 1. Introduction

Global climatic change due to increasing greenhouse gas emissions has induced drought and/or higher temperatures in recent decades [1]. Seed germination and seedling establishment are both influenced by rising temperature and moisture limitation, making these life stages particularly sensitive to climate change [1–5]. Most species of the genus *Lysimachia* are perennial herbaceous plants, mainly distributed in the temperate zones of the Northern Hemisphere and the tropical regions of Southeast Asia and southwest China. *Lysimachia coreana* Nakai is an important wild plant endemic to Korea. It is classified as a least-concern (LC) species and has been designated as a species vulnerable to climate change by the Korea Forest Service. Nevertheless, there are only a few reports on the seed morphology and taxonomy of *L. coreana* [6] and no studies on seed physiology.

Seed dormancy is an intrinsic characteristic of seeds that determines the conditions under which seeds can germinate. Genetics, environment, and temperature have profound impacts on seed dormancy establishment and release. DOG1 (delay of germination 1) has been discovered as a key player in the control of seed dormancy establishment downstream of the temperature pathway, where it is transformed into internal signals, eventually leading to changes in the physiological processes in seeds [7], and plant hormones such as abscisic acid and gibberellins (GAs), also play a role, at least in part, in influencing seed dormancy [8,9]. Abscisic acid has been linked to seed dormancy induction and

maintenance, whereas GA is linked to dormancy breaking and germination [10,11]. In addition, cold stratification (CS) is an important environmental element that alleviates seed dormancy [12,13]. Depending on the species, the effective temperature for seed dormancy release during CS ranges from 0 to 10 °C [13–15].

Seed dormancy mechanisms were classified into five categories by Baskin and Baskin [16]: physical dormancy (PY) refers to seeds with a hard water-impermeable seed coat, physiological dormancy (PD) refers to seeds with a physiological inhibitory mechanism that prevents radicle emergence, morphological dormancy (MD) refers to seeds that have a small or undeveloped embryo, morphological dormancy (MPD) refers to seeds that have PD but also have an underdeveloped embryo that needs time to grow before germination, and combinational dormancy (PY + PD) refers to seeds possessing both PY and PD. *Lysimachia davurica* seeds germinated at 30 °C and had 94.7% GR after 200 mg·L$^{-1}$ GA$_3$ treatment [17]. Among seeds belonging to the same genus, *Lysimachia vulgaris* L. also were confirmed to have PD [18]. Generally, seed dormancy is a species-specific mechanism, but even within the same classification class (family and genus), the types and degrees of species vary depending on the geographic distribution and ecological characteristics of the target species [19]. There is still insufficient knowledge of seed dormancy in *L. coreana* Nakai.

The Baekdudaegan Global Seed Vault (BGSV) was established for ex situ conservation against plant extinction, particularly for wild plant species. The seed vault is a facility for the permanent storage of seeds, unlike seed banks, which allows for the storage and use of seeds. The permanent storage of seeds preserves biodiversity and mitigates extinction risks caused by climate change and natural disasters. Therefore, we investigated the optimal germination conditions of seeds for permanent storage in the BGSV and analyzed the dormancy and dormancy breaking methods to obtain basic data for wild plant conservation and propagation in preparation for future global climate change.

## 2. Materials and Methods

### 2.1. Seed Material and Collection

Seeds of *L. coreana* were collected from plants growing in Bonghwa-gun, Gyeongsangbuk-do, Korea, in 2017 and dried for 90 d in the drying room of the Baekdudaegan National Arboretum (15 °C, 15% humidity). The dried seeds were sealed in a glass bottle and stored in a seed bank at −20 °C and 40% humidity until they were required for the experiments.

### 2.2. Morphological Characteristics of the Seeds

Ten seeds were randomly selected, and their seed characteristics, including shape, size, and embryo structure, were measured. The seeds were cut into thin sections using a razor blade, and the embryo structure was measured using a digital microscope (DVM6 PlanApo FOV 3.60; Leica Microsystems, Wetzlar, Germany). A Tetrazolium (TZ) test analyzed the possibility of identifying *L. coreana* seed vigor or dormancy [20]. The ratio of embryo length to seed length (E:S ratio) was calculated using the formula detailed by Vandellook et al. [21].

$$\text{E:S ratio = seed length/embryo length}$$

### 2.3. Effects of Temperature Treatment on Seed Germination

The effect of temperature was tested at four constant (5, 15, 20, and 25 °C) and two alternating (20/10 and 25/15 °C) temperature treatments. At each temperature, seeds were incubated under a 12 h/12 h (light/dark) regime in a growth chamber (TGC-130H, Espec Mic Corp., Aichi, Japan). All of the temperature conditions were established by reflecting the four seasons of Korea. For each experiment, five replicates of 10 seeds (in Petri dishes with 1% agar medium) were incubated at each temperature for 40 d. Radicle emergence was monitored daily to calculate the percentage of germination. A seed was considered "germinated" when the radicle reached at least 2 mm in length [22].

*2.4. Effects of GA₃ Treatment and Temperature Treatment on Germination*

To assess the influence of gibberellic acid (GA$_3$), five replicates of 10 seeds were placed in Petri dishes with 1% agar containing 0 (only 1% agar medium), 100, 250, and 500 mg·L$^{-1}$ GA$_3$ stock solution ($\geq$90%, Sigma-Aldrich, St. Louis, MO, USA) [23]. They were then incubated at four constant (5, 15, 20, and 25 °C) and two alternating (20/10 and 25/15 °C) temperatures under a 12 h/12 h (light/dark) regime for 30 d. Radicle emergence was monitored daily to calculate the percentage of germination. A seed was considered "germinated" when the radicle reached at least 2 mm in length.

*2.5. Effects of Warm and/or Cold Stratification Periods on Germination*

The seeds were stratified at 5 °C and 12 h/12 h (light/dark) or 25/15 °C and 12 h/12 h (light/dark) for 0, 6, and 9 weeks. The seeds were placed on two sheets of filter paper (Whatman No. 2) in 90 × 15 mm Petri dishes and moistened with distilled water during the stratification treatments. Following each stratification period, five replicates of 10 seeds (in Petri dishes with 1% agar medium) were incubated at four constant (5, 15, 20, and 25 °C) and two alternating (20/10 and 25/15 °C) temperatures under a 12 h/12 h (light/dark) regime for 30 d. Radicle emergence was monitored daily to calculate the percentage of germination. A seed was considered "germinated" when the radicle reached at least 2 mm in length.

*2.6. Germination Characteristics Analysis*

At the end of the germination period, the germination rate (GR) according to the Association of Official Seed Analysts (AOSA) method [22], the time to achieve a 50% germination rate (T$_{50}$) [24], and the germination index (GI) which is a measure of the percentage and speed of germination [25] were calculated according to the following formulas:

$$GR\ (\%) = (N/S) \times 100$$

$$T_{50}\ (days) = T_i + [(N/2) - N_i\} * (T_j - T_i)]/(N_j - N_i),$$

$$GI = \sum(T_x \cdot N_x)/S$$

where $N$ is the total number of germinations, $S$ is the total number of seeds, $N_i$ and $N_j$ are the cumulative number of seeds germinated by adjacent counts at times $T_i$ and $T_j$, respectively, when $N_i < N/2 < N_j$, $T_x$ is the number of days after seed planting, and $N_x$ is the number of germinations on the day of investigation.

*2.7. Statistical Analyses*

Statistical analysis was performed using the Statistical Package for Social Science version 26.0 (SPSS, Chicago, IL, USA). The analysis of variation among groups was conducted using a two-way ANOVA, and the experiments with GA$_3$ and stratification treatments were analyzed using Tukey's honestly significant difference test for GR and Scheffe's test for T$_{50}$ and GI where there are no result, so the number of repetition is different. In addition, the interaction between the main effects of temperature and GA$_3$ or temperature and stratification was analyzed using multivariate analysis of variance (MANOVA) ($p < 0.05$).

## 3. Results

### 3.1. Morphological Characteristics of Seeds

The seeds had various shapes, and their average size was about 1.15 ± 0.03 mm in width and 0.81 ± 0.03 mm in length. The seed was dark brown with a rough texture. The embryo structure was assumed to be a miniature axile embryo. The E:S (embryo/seed length) ratio was 0.84 ± 0.04. The tetrazolium test (TZ) [25] was used to check seed vigor, which revealed a red-colored embryo that confirmed germination ability (Figure 1).

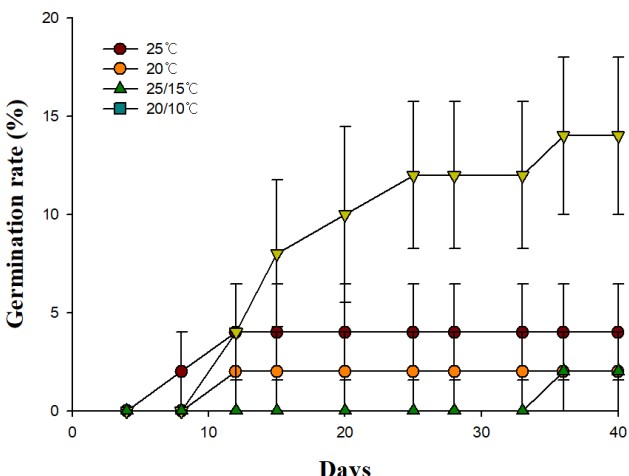

**Figure 1.** Morphology of a *Lysimachia coreana* seed. The left image shows the texture and morphological characteristics, and the right image shows the cross-section of the seed with the tetrazolium-stained embryo structure. Images were taken using a digital microscope. Scale bars: 1 mm.

*3.2. Effects of Temperature Treatment on Seed Germination*

The germination of *L. coreana* seeds varied significantly with the temperature. Among all treatments, the final GR was the highest at 14.0% at 25/15 °C, while no germination was observed at 5 and 15 °C (Figure 2). Only 4.0% and 2.0% of the seeds germinated at 25 °C and 20 °C and 20/10 °C, respectively, after 40 d of incubation. Most seeds began to germinate between 5 and 10 d and showed a greater tendency to germinate at higher temperatures than at lower temperatures. According to a previous study, we found that germination did not occur at 25 °C and 12 h/12 h (light/dark) conditions, whereas in the tetrazolium test, is a rapid method for evaluation of seed viability (Figure 1), it was found that *L. coreana* had viable seeds but a low GR for various temperature treatments. Therefore, they may have been dormant.

**Figure 2.** Effect of temperature treatments on the cumulative germination rate (%) in seeds of *L. coreana*. Bars indicate standard error (*n* = 5). It shows the germinated temperature treatments, excluding the non-germinated (5 and 15 °C).

*3.3. Effects of GA₃ Treatment and Temperature Treatment on Germination*

Under all temperature treatments, except at 5 °C, GR was mostly enhanced by the 100 and 250 mg·L⁻¹ GA₃ treatments. On comparing the difference in germination tendencies by GA₃ concentration, it was found that 250 mg·L⁻¹ GA₃ treatment had the highest germination tendencies at various temperature treatments (Table 1). In particular, with 250 mg·L⁻¹ GA₃ treatment, the highest GR of 82.0% was obtained at 25/15 °C, followed by 76.0 and 66.0% at 20/10 and 25 °C, respectively. The $T_{50}$ values were 13.3 (25/15 °C), 20.9% (20/10 °C), and 10.8 (25 °C) d, which were shorter than those of the other experimental

groups but not statistically significant. The GI values were 13.3, 16.3, and 8.1 at 25/15, 20/10, and 25 °C with 250 mg·L$^{-1}$ GA$_3$, respectively. Compared to other treatments, the GI obtained by dividing the number of germinations on investigation days by the total number of occupations was higher than that in the other experimental groups. In addition, the correlation between temperature and GA$_3$ factors was analyzed using MANOVA. In the case of GR, a significant correlation was observed between all treatment factors. T$_{50}$ and GI showed a high correlation with GA$_3$ ($p < 0.001$), while T$_{50}$ was not correlated with T × GA$_3$, and GI showed a low level of correlation with temperature ($p < 0.05$) (Table 1).

**Table 1.** The effect of gibberellic acid treatments (100, 250, and 500 mg·L$^{-1}$) and temperature treatments on the germination vigor of *L. coreana* seeds.

| T (°C) | GA$_3$ (mg·L$^{-1}$) | GR (%) | T$_{50}$ (Days) | GI |
|---|---|---|---|---|
| 25 | Control | 4.0 ± 2.5 [h] | 10.3 ± 3.0 [ns] | 0.4 ± 0.3 [d] |
| | 100 | 38.0 ± 6.6 [def] | 19.5 ± 6.4 [ns] | 6.9 ± 1.9 [abcd] |
| | 250 | 66.0 ± 4.0 [abc] | 10.8 ± 1.6 [ns] | 8.1 ± 1.1 [abcd] |
| | 500 | 30.0 ± 12.3 [defg] | 12.6 ± 1.1 [ns] | 3.7 ± 1.4 [bcd] |
| 20 | Control | 2.0 ± 2.0 [h] | - | 0.2 ± 0.2 [d] |
| | 100 | 24.0 ± 5.1 [efgh] | 16.7 ± 3.3 [ns] | 5.0 ± 1.5 [bcd] |
| | 250 | 52.0 ± 2.0 [bcd] | 20.2 ± 4.8 [ns] | 10.8 ± 1.6 [abcd] |
| | 500 | 48.0 ± 5.8 [cde] | 20.0 ± 0.9 [ns] | 9.9 ± 1.6 [abcd] |
| 15 | Control | 0.0 ± 0.0 [h] | - | - |
| | 100 | 16.0 ± 8.1 [fgh] | 27.5 ± 0.9 [ns] | 4.4 ± 2.3 [abcd] |
| | 250 | 36.0 ± 9.8 [defg] | 24.0 ± 1.5 [ns] | 8.8 ± 2.5 [abcd] |
| | 500 | 48.0 ± 3.7 [cde] | 20.5 ± 1.8 [ns] | 11.5 ± 0.8 [abc] |
| 25/15 | Control | 14.0 ± 4.0 [fgh] | 16.1 ± 2.3 [ns] | 2.6 ± 0.8 [bcd] |
| | 100 | 32.0 ± 2.0 [defg] | 28.4 ± 2.5 [ns] | 8.5 ± 0.8 [abcd] |
| | 250 | 82.0 ± 3.7 [a] | 13.3 ± 0.9 [ns] | 13.3 ± 1.1 [ab] |
| | 500 | 12.0 ± 3.7 [gh] | 18.8 ± 3.2 [ns] | 2.2 ± 0.7 [cd] |
| 20/10 | Control | 2.0 ± 2.0 [h] | - | 0.7 ± 0.7 [d] |
| | 100 | 32.0 ± 2.0 [defg] | 28.4 ± 2.5 [ns] | 8.5 ± 0.8 [abcd] |
| | 250 | 76.0 ± 5.1 [ab] | 20.9 ± 0.9 [ns] | 16.3 ± 0.8 [a] |
| | 500 | 16.0 ± 5.1 [fgh] | 21.5 ± 4.9 [ns] | 4.2 ± 1.6 [bcd] |
| 5 | Control | 0.0 ± 0.0 [h] | - | - |
| | 100 | 0.0 ± 0.0 [h] | - | - |
| | 250 | 0.0 ± 0.0 [h] | - | - |
| | 500 | 2.0 ± 2.0 [h] | 36.0 ± 0.0 [ns] | 0.7 ± 0.7 [d] |
| *p*-value | T [1] | ** | *** | * |
| | GA$_3$ | *** | *** | *** |
| | T × GA$_3$ | *** | ns | *** |

[1] T, temperature. The germination rate (GR) was analyzed by Tukey's honestly significant difference (HSD) test, and the time to achieve a 50% germination rate (T$_{50}$) and germination index (GI) were analyzed using Scheffe's test. Values are presented as the mean ± standard error ($n = 5$). The mean values sharing the same letters do not differ significantly. Data with different lower case superscript letters in the same column indicate that they are statistically different ($p < 0.05$). Asterisks represent statistically significant differences in the interaction between temperature and gibberellic acid (GA$_3$) according to multivariate analysis of variance (MANOVA) (* $p < 0.05$; ** $p < 0.01$; *** $p < 0.001$; ns, not significant).

### 3.4. Effects of Warm and/or Cold Stratification Periods on Germination

Warm and/or cold stratification had a significant effect on dormancy break and germination (Table 2). Under WS, the seeds showed a low germination tendency under all temperature treatments (Table S1). However, the GR was improved by CS. In particular, the highest GR of seeds was 56% after CS for 9 weeks, followed by incubation at 25/15 °C with a faster T$_{50}$ (3.9 d) and a higher GI (3.0) than the other experimental groups. The seeds also showed high germination rates of 42, 52, and 48% at 25 °C, 20, and 20/10 °C, respectively,

after 9 weeks of CS. Comparing the correlation between temperature and CS and/or WS treatments, a significant correlation was found between germination abilities (GR and $T_{50}$) and temperature and stratification factors ($p < 0.001$). The GI showed a strong ($p < 0.001$) correlation with temperature, but no correlation was observed with stratification; GR was also not correlated with T $\times$ S (Table 2).

**Table 2.** The effect of cold (5 °C) stratification and temperature treatments (for 6 or 9 weeks) on the germination vigor of *L. coreana* seeds.

| T (°C) | S (week) | GR (%) | $T_{50}$ (days) | GI |
|---|---|---|---|---|
| 25 | CS-6 | 16.0 ± 6.8 [cd] | 2.4 ± 0.4 [a] | 0.4 ± 0.2 [b] |
| | CS-9 | 42.0 ± 8.6 [abcd] | 2.9 ± 0.2 [ab] | 1.4 ± 0.3 [ab] |
| 20 | CS-6 | 32.0 ± 3.7 [abcd] | 4.4 ± 0.7 [ab] | 2.4 ± 0.7 [ab] |
| | CS-9 | 52.0 ± 5.6 [ab] | 3.7 ± 0.5 [ab] | 2.5 ± 0.6 [ab] |
| 15 | CS-6 | 12.0 ± 3.7 [d] | 3.5 ± 0.4 [ab] | 0.5 ± 0.2 [b] |
| | CS-9 | 22.0 ± 3.7 [bcd] | 3.8 ± 0.3 [ab] | 0.9 ± 0.2 [b] |
| 25/15 | CS-6 | 46.0 ± 9.3 [abc] | 4.4 ± 0.5 [ab] | 2.5 ± 0.7 [ab] |
| | CS-9 | 56.0 ± 4.0 [a] | 3.9 ± 0.4 [ab] | 3.0 ± 0.6 [ab] |
| 20/10 | CS-6 | 36.0 ± 2.5 [abcd] | 8.1 ± 0.4 [bc] | 3.6 ± 0.6 [ab] |
| | CS-9 | 48.0 ± 8.6 [ab] | 3.8 ± 0.9 [ab] | 2.4 ± 0.7 [ab] |
| 5 | CS-6 | 24.0 ± 5.1 [bcd] | 24.0 ± 1.0 [d] | 5.8 ± 1.3 [a] |
| | CS-9 | 16.0 ± 6.0 [cd] | 9.9 ± 2.3 [c] | 1.6 ± 0.7 [ab] |
| *p*-value | T [1] | *** | *** | *** |
| | S [2] | *** | *** | Ns |
| | T × S | ns | *** | ** |

[1] T, temperature; [2] S, stratification. The germination rate (GR) was analyzed by Tukey's honestly significant difference (HSD) test, and the time to achieve a 50% germination rate ($T_{50}$) and germination index (GI) were analyzed using Scheffe's test. Values are presented as the mean ± standard error (*n* = 5). The mean values sharing the same letters did not differ significantly. Data with different lower case superscript letters in the same column indicate that they are statistically different ($p < 0.05$). Asterisks represent statistically significant differences in the interaction between temperature and stratification according to multivariate analysis of variance (MANOVA) (** $p < 0.01$; *** $p < 0.001$; ns: not significant).

### 3.5. Comparison of Seed Germination According to $Ga_3$ and Stratification under Optimal Germination Conditions

Based on the results obtained from the above three experiments, the optimal germination temperature of *L. coreana* seeds was 25/15 °C and 12 h/12 h (light/dark). Under the optimal germination temperature condition of 25/15 °C, *L. coreana* seeds that were treated with 250 mg·L$^{-1}$ GA$_3$ and cold stratification (CS-9) treatments showed differences in GR of 82 and 56%, respectively. It was found that the WS treatment had no effect on seed dormancy breaking compared to the untreated control (Figure 3). The experiment on the uniformity of the germination period ($T_{50}$) showed the lowest $T_{50}$ values of 3.9, 4.4, and 13.3 d for CS-9, 6, and GA$_3$ 250 mg·L$^{-1}$ treatments, respectively. Likewise, the highest GI was 13 in GA$_3$ 250 mg·L$^{-1}$, followed by 8 in GA$_3$ 100 mg·L$^{-1}$ (Figure 3).

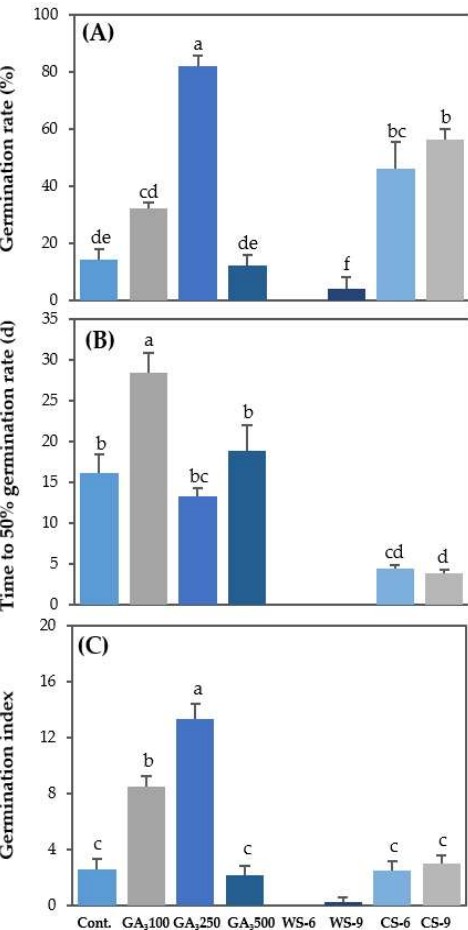

**Figure 3.** Comparison of GA$_3$ and stratification (CS and WS) treatments on the germination vigor of *L. coreana* seeds under optimum germination conditions (25/15 °C, 12 /12 h [light/dark]). (**A**) Germination rate (GR, %); (**B**) the time to achieve a 50% germination rate (T$_{50}$, days); (**C**) Germination index (GI). Data were analyzed by Tukey's honestly significant difference test. Bars indicate the standard error (*n* = 5). Mean values sharing the same letters do not differ significantly (*p* < 0.05).

## 4. Discussion

*L. coreana* is a rare (least concern: LC [26]), endemic, and climate change vulnerable species. It mainly grows in sunny wetlands or near streams, and its flowering period is from June to July. The shape of its seeds varies, the average size of seeds was 1.15 × 0.81 mm (length × width), and the structure of the embryo was assumed to be a fully developed miniature axile shape. Generally, seeds with MD or MPD have small and underdeveloped embryos [16,27,28] and the ratio of E:S is 0.5 or less [29]. We confirmed that the E:S ratio in *L. coreana* seeds was 0.84 ± 0.04. Therefore, *L. coreana* seeds did not belong to the MD or MPD dormancy types (Figure 1). In a previous study, the seeds did not germinate at 25 °C, 12 h/12 h (light/dark) conditions, and TZ staining showed that the seed coat was water-permeable by staining the embryo red, which confirmed seed viability (Figure 1) and led to their classification as dormant seeds. Generally, seeds with PY have impermeable seed coats [16,30]. However, in our study, all *L. coreana* seeds absorbed water and were stained in the TZ test, indicating that they did not have PY. Based on the above observations, we postulate that *L. coreana* seeds have PD because these seeds are water-permeable and radicle emergence is prevented by a physiological inhibitory mechanism in the embryo [31–33]. Baskin and Baskin [16] categorized PD into three categories: non-deep, intermediate, and deep. At non-deep levels of PD, GAs promote seed germination, and CS (0–10 °C) or WS (>15 °C) alleviates seed dormancy. In the intermediate level of PD, some species are effective in promoting germination with GAs; moreover, these seeds require 2–3 months of

CS to break their dormancy. Since *L. coreana* seeds showed enhanced germination by GAs and the dormancy was broken after CS (Tables 1 and 2), we could rule out deep levels of PD in these species, in which GAs do not promote germination and seeds require 3–4 months of CS to break dormancy [16,33].

Laboratory experiments were conducted under various temperature treatments to determine the optimal germination temperature for *L. coreana* seeds. Temperature is an important environmental factor that regulates seed dormancy induction and expression throughout seed development and germination [34–40]. Moreover, temperature gradients are related to both the dormancy depth and germination requirements of non-dormant seeds [41–44]. In many species, seed germination is promoted more effectively by alternate temperatures rather than at constant temperatures [45,46]. In our study, *L. coreana* seeds had the highest GR of 14.0% at 25/15 °C and 12 h/12 h (light/dark) for 40 d of incubation, although the GR was very low (Figure 2). Furthermore, under almost all temperature treatments, seeds generally showed a low germination tendency or rarely germinated, which suggests that they may possess a physiological inhibiting mechanism that prevents radicle emergence. Especially, the difficulty of dormant seeds to sprout when placed at ideal temperatures appears to be linked to abscisic acid generation during imbibition [47–52].

Stratification (cold or warm) requirements for dormancy breaking and seed response to GA pretreatment are crucial considerations when attempting to determine between non-deep and intermediate PD. $GA_3$ pretreatment is known to promote the germination of seeds with non-deep PD, but this is not the case for all species with intermediate PD [16,53]. $GA_3$ enhances non-dormant seed germination and germination maintenance, it has also been observed in numerous other plant species [54–57]. The GR of *L. coreana* increased from 14.0% (control) to 82.0% at 25/15 °C with 250 mg·$L^{-1}$ $GA_3$ treatment, and improved 76.0% and 66.0% at 20/10 and 25 °C with 250 mg·$L^{-1}$ $GA_3$ treatment, respectively. However, under 500 mg·$L^{-1}$ $GA_3$, the germination rate tended to decrease under 25, 20, 25/15, 20/10 °C temperature treatments (Table 1). $GA_3$ pretreatment might not enhance germination in any of the seeds and may even prevent germination depending on the technique and quantity [53]. For example, *L. coreana* seeds whose germination was promoted by 250 mg·$L^{-1}$ $GA_3$ were found to have greater germination at higher temperatures than at lower temperatures. On the other side, *Oenanthe stolonifera* overcame seed dormancy but, was not dramatically improved the GR by $GA_3$ treatment [58]. Therefore, the proper treatment, concentration, and other circumstances of each species must be investigated [59].

Non-deep PD does not require an exact period of stratification (cold or warm). Specifically, short periods (a few days to a few weeks) of stratification (cold or warm) can be used to break non-deep PD species. [33]. The *L. coreana* seeds were shown to undergo dormancy breaking under low-temperature conditions for 6 and 9 weeks. After 9 weeks of CS at 5 °C, seeds incubated at 25/15 °C showed the greatest GR of 56 % and the next highest GR of 52, 48 % at 20, 20/10 °C after 9 weeks of CS, respectively. Whereas those that underwent WS at 25/15 °C did not germinate or had very low germination rates under most incubation condition (Table 1, S1). Therefore, dormancy type of *L. coreana* seeds are closer to non-deep PD than intermediate PD (require CS treatment with 2–3 months) [29]. The CS of seeds naturally occurred when they are exposed to consistent temperatures of 0–1 °C in snow-covered soil over the winter. Although we chose a higher temperature (5 °C) for our CS treatments, stratification temperatures of 3–5 °C are widely used in seed germination studies because they break dormancy in many different species [19,47]. Seed germination in various plant species requires a period of CS. Stratification induces the activation of GA biosynthesis in intact seeds for germination [60]. Many studies suggest that CS is the most effective treatment for breaking dormancy in the seeds of *Ferula gummosa*, *Thalictrum mirabile*, and *Isatis violascens* [61–63].

In this study, the *L. coreana* seeds germinated after $GA_3$ treatment and broke dormancy after CS treatment (Tables 1 and 2). Upon comparing the germination characteristics between $GA_3$ and stratification treatments under the optimal germination temperature condition at 25/15 °C, the germination rates of *L. coreana* seeds in 250 mg·$L^{-1}$ $GA_3$ and CS-9

treatments were 82 and 56%, respectively, with a low $T_{50}$ (Figure 3). These results indicate that the *L. coreana* seeds break dormancy through $GA_3$ treatment or CS, but treatment with 250 mg·L$^{-1}$ $GA_3$ treatment was the most effective method. $GA_3$ treatments could be a substitute for stratification or a portion of the stratification period [64,65]. In addition, the seeds germinated at high temperatures rather than at low temperatures, especially when the optimal germination temperature was 25/15 °C. Rossello and Mayol [66] reported that *L. minoricensis* showed high germination under most temperature conditions, with the exception of those performed at low temperatures, and L. mauritiana seeds showed high germination at 25 °C in light [67]. Therefore, the germination characteristics of *L. coreana* seeds reflect the phenology of germination in the spring following the previous year's flowering in June–July and cold winter temperatures. Seed dormancy and germination requirements are key characteristics that determine the ecological and geographic distribution of plants [53,61].

## 5. Conclusions

Based on the findings of this study, it can be concluded that seed dormancy in *L. coreana* may be categorized as a non-deep PD type, and $GA_3$ or CS at 25/15 °C is required to break dormancy and promote germination. In particular, 250 mg·L$^{-1}$ $GA_3$ treatment was more effective for the promotion of seed germination than CS for 9 weeks, and *L. coreana* seeds preferred a high temperature for germination. Therefore, the optimum germination temperature for *L. coreana* seeds is an alternative temperature of 25/15 °C. Elucidating the type of seed dormancy, dormancy breaking methods, and optimum germination conditions for individual species are necessary to protect the genetic diversity of endangered plant species by improving seed storage and plant propagation methods.

**Supplementary Materials:** The following are available online at https://www.mdpi.com/article/10.3390/horticulturae7110490/s1, Table S1: Warm (25/15 °C) stratification and temperature treatments (for 6 or 9 weeks) on the germination vigor.

**Author Contributions:** Conceptualization, C.S.N.; methodology, S.G.B.; validation, S.Y.W. and M.J.K.; formal analysis, S.G.B.; investigation, S.G.B.; resources, C.H.P. and M.H.L.; data curation, S.G.B. and M.J.K.; writing—original draft preparation, S.G.B. and J.H.I.; writing—review and editing, S.G.B. and M.J.K.; visualization, S.G.B.; supervision, S.Y.W. and C.S.N.; project administration, C.S.N.; funding acquisition, C.S.N. All authors have read and agreed to the published version of the manuscript.

**Funding:** This study was supported by the R&D Program for Forest Science Technology (Project No. 2021400B10-2125-CA02) provided by the Korea Forest Service.

**Institutional Review Board Statement:** Not applicable.

**Informed Consent Statement:** Not applicable.

**Data Availability Statement:** The data presented in this study are available upon request from the corresponding author. The data are not publicly available because of privacy or other restrictions.

**Acknowledgments:** We would like to thank the biological resources research team in the National Arboretum Baekdudaegan for seed collections.

**Conflicts of Interest:** The authors declare no conflict of interest. The funders had no role in the study design, collection, analyses, or interpretation of data; in the writing of the manuscript; or in the decision to publish the results.

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
