# Peer review of "Non-Deep Physiological Dormancy in Seed and Germination Requirements of Lysimachia coreana Nakai"

_horticulturae, doi:10.3390/horticulturae7110490_

Round 1

Reviewer 1 Report

This manuscript entitled " Non-Deep Physiological Dormancy in Seed and Germination Requirements of Lysimachia coreana Nakai" investigated the optimal germination conditions of seeds for permanent storage and analyzed the dormancy and dormancy breaking methods in Lysimachia coreana Nakai. However, I still have the following concerns to be revised by the author.

Concerns:

  1. The sample size (i.e., five replicates of 10 seeds) used for different experiments was too small. Therefore, the Standard Error (Table 1, 2 and Fig. 3) is too high in each sample set. So, the sample size should be increase (i.e., five replicates of 50-100 seeds) for precise results.
  2. The effect of Osmotic priming agents (such as KNO3, KCl, and NH4NO3) should also be checked in this study.
  3. Several recent and important papers dealing with non-deep physiological dormancy in seeds in different species were not cited in the manuscript. Two of them is given below:

Kim, H.-M.; Kim, J.-H.; Lee,D.-H.; Jung, Y.-H.; Park, C.-Y.; Lee, M.-H.; Kim, K.-M.; Lee, J.-H.; Na, C.-S. Non-Deep Simple Morphophysiological Dormancy and Germination Characteristics of Gentiana triflora var. japonica (Kusn.) H. Hara (Gentianaceae), a Rare Perennial Herb in Korea. Plants 2021, 10, 1979. https://doi.org/10.3390/plants10101979

Oh, H.J., Shin, U.S., Lee, S.Y. et al. Non-deep physiological dormancy in seeds of Euphorbia jolkinii Boiss. native to Korea. j ecology environ 45, 20 (2021). https://doi.org/10.1186/s41610-021-00194-x

  1. The discussion section is poorly written and not well connected to the results. It seems to be just repetitions of results. So, I suggested that the authors should improve the discussion section. They have to discuss their results and compare them with some earlier and recently published papers in more depth and clarity.

Minor issues

  1. In some places (Line 35, 79, 90, 145, 149, 161, 179, 226) authors used “Lysimachia coreana” and at some places (38, 68, 157, 210, 231) authors used “L. coreana” randomly, In my opinion, after initial use of full form abbreviation should be used elsewhere.
  2. In Table 1 and Table 2, please provide full abbreviations or meaning for h, efgh, abcd etc.
  3. In Figure 4, use either (a) or (A) to make figure and legend uniform.

Author Response

Dear Editor,

Title

Non-Deep Physiological Dormancy in Seed and Germination Requirements of Lysimachia coreana Nakai

We are pleased to resubmit for publication the revised version of entitled “Non-deep Physiological Dormancy in Seed and Germination Requirements of Lysimachia coreana Nakai”. We are very grateful to the Associate Editor and the two Reviewers for their deep and detailed comments which have helped us to improve on our manuscript. We have revised our manuscript according to their suggestions and comments of two anonymous reviewers. We have carefully addressed comments by each reviewer as outlined below. We hope that the reviewers and the editors will be satisfied with author's reply to the review report and the revised version.

Sincerely yours,

Reviewer 2 Report

This is a review of manuscript “Non-Deep Physiological Dormancy in Seed and Germination Requirements of Lysimachia coreana Nakai" submitted to Journal: Horticulturae.

The article may be of interest to breeders and scientists studying the physiological basis of seed germination.

Title is informative and dense enough.

Abstract fully reflects the content of the manuscript. 

In my opinion the manuscript needs to be improved.

Main suggestions:

  1. It is preferably to exclude repetition of words in the title and «keywords»
  2. The phrase "temperature regime" would be better replaced with "temperature treatment”.
  3. It would be better to add explanatory information: what was the reason for the choice of germination temperature, temperature of alternating modes?

Why was the temperature step of 100C chosen (in alternating modes)?

  1. What is the reasoning behind the choice of gibberellic acid concentration? Why the different step in concentrations?
  2. Why were only two stratification temperatures chosen?
  3. Why were the stratification time intervals chosen in different steps?
  4. Figure 3 is hard to see. Change the type of data presentation so that the graphs do not overlap with each other. The graph showing the values of germination at 150C is not visible. Why is the standard error the same everywhere on the graph, while the table shows different data?

  1. It is recommended to indicate HSD data in the tables.

  1. In the experiment, it would be necessary to increase the gibberellic acid concentration variants, especially in the case of 50C and 150C temperatures, to trace a further increase in GR.

In this case, the conclusions would be more reasonable.

  1. L 163- «..Under all temperature regimes, except at 5 °C, GR was mostly enhanced by the 100, 250, and 500 mg·L–1 GA3 treatments.» - The conclusion is not correct: not all temperature treatments at a concentration of 500 mg/L gibberellic acid resulted in an increase in GR. This statement is true only for concentrations of 100 and 250 mg/L GR.

  1. it is recommended to calculate the share of influence of each factor in multivariate analysis.

  1. What does the label in Figure 4(B) mean: "d" - days ?
  2. It is desirable to remove duplicative information from the literature review from the discussion and to make it more concise and laconic.

Author Response

(The authors gave the same response as above.)

Round 2

Reviewer 1 Report

The authors have clarified most of the questions I raised in my previous review. Now I feel the manuscript could be accepted for publication.